# Thermal Pyrolysis of Polystyrene Aided by a Nitroxide End-Functionality Improved Process and Modeling of the Full Molecular Weight Distribution

**DOI:** 10.3390/polym14010160

**Published:** 2021-12-31

**Authors:** Antonio Monroy-Alonso, Almendra Ordaz-Quintero, Jorge C. Ramirez, Enrique Saldívar-Guerra

**Affiliations:** Centro de Investigación en Química Aplicada, Blvd. Enrique Reyna 140, Saltillo 25192, Mexico; antonio.monroy.d21@ciqa.edu.mx (A.M.-A.); almendra.ordaz@ciqa.edu.mx (A.O.-Q.); jorge.ramirez@ciqa.edu.mx (J.C.R.)

**Keywords:** polystyrene pyrolysis, mathematical modeling, molecular weight distribution

## Abstract

A significantly improved thermal pyrolysis process for polystyrene (PS) is reported and mathematically modeled, including the description of the time evolution of the full molecular weight distribution of the polymer during its degradation by direct integration of the balance equations without simplifications. The process improves the styrene yield from 28–39%, reached in our previous report, to 58–75% by optimizing the heating ramp during the initial stage of the pyrolysis process. The process was tested at 390 and 420 °C on samples of conventional PS synthesized via free-radical polymerization (FRP) and PS with a nitroxide end-functionality synthesized via nitroxide mediated polymerization (NMP) with three levels of the nitroxide to initiator (N/I) molar ratio: 0.9, 1.1 and 1.3. The NMP-PS produced with N/I = 1.3 generates the highest styrene yield (75.2 ± 6.7%) with respect to the best FRP-PS yield (64.9 ± 1.2%), confirming the trends observed in our previous study. The mathematical model corrects some problems of a previous model that was based on assumptions that led to significant errors in the predictions; this is achieved by solving the full molecular weight distribution (MWD) without assumptions. The model provides further insight into the initial stages of the pyrolysis process which seem to be crucial to determine the chemical paths of the process and the styrene yield, as well as the influences of the initial heating ramp used and the presence of a nitroxide end-functionality in the polymer.

## 1. Introduction

The outstanding physical and chemical properties of plastics which have led to their excessive production and consumption are nowadays causing severe irreversible damage to the environment.

Amongst the variety of potential solutions for waste management that include recycling, landfill disposal and combustion, pyrolysis is the one technique that allows the recovery of high-value products usually in the form of complex mixtures. This alternative has acquired significant industrial and academic interest in recent years as a possible and promising disposal option; it does not require dumping waste in landfills, toxic gas production is minimized, and valuable products are recovered, including fuels and monomers that can be later used in other processes or in the synthesis of new polymers [1,2,3].

Pyrolysis can be carried out via catalytic or thermal routes, and derived from the fact that it is strongly influenced by a wide range of process parameters it can also be finely tuned to favor one specific product distribution over others. These operation parameters include the composition of the treated waste, reaction temperature, pressure, presence or absence of oxygen, heating rate, moist content, residence time, and presence of catalysts or solvents; the set of selected pyrolysis parameters must be strictly controlled during the whole reaction process as they will directly dictate its course and hence, the spectrum and yield of the formed species.

In this work, we report a significantly improved process for the thermal pyrolysis of PS based on a process previously published by our group [1]. In the present process, the yields of styrene monomer recovered increase from about 28–38% to about 58–75%. Moreover, a more detailed and enhanced mathematical model for the description of the process is described and discussed, based on a previous model developed in our group [1]. In the new model, assumptions used in our previous model are removed and are proved to generate serious deviations from what the new assumption-free model predicts.

The pyrolysis process described in the present work sticks out as it allows the recovery of large amounts of styrene monomer despite using a purely thermal approach; the selected pyrolysis temperatures are slightly higher than the ceiling temperature of polystyrene (PS). High selectivity of products is achieved (similar to that of a catalytic process, without the burden of dealing with catalysts) by optimizing the initial heating ramp in the process and enhanced even further by introducing a nitroxide moiety at the end of the PS chains. Before describing our improved process as well as the enhanced mathematical model of the polystyrene pyrolysis process and its supporting empirical evidence, a brief state of the art is given. First, an examination of the polystyrene pyrolysis reaction mechanism is provided, then a review of the developed work on identifying the PS pyrolysis products, followed by a brief inspection of the nitroxide polymerization and how it can influence and improve the pyrolysis process by inducing depolymerization, and lastly, a quick examination of the use of a mathematical model as an aid to understand and in some way predict this complex process.

### 1.1. Background: Pyrolysis Mechanism

Pyrolysis of polymer waste is not a new topic, it has been extensively documented that multiple and complex reactions take place on the polymer structure by means of heat. Usually, at low pyrolysis temperatures, products are obtained predominantly in the form of a liquid mixture and as the process temperature is increased, gases are recovered; meanwhile, solids are reclaimed at prolonged reaction times [4]. When implementing a purely thermal pyrolysis process, operation temperatures can reach up to 700 °C with poor control over the yield or type of products obtained, contrary to the catalytic pyrolysis, which ensures the selectivity of the products at lower temperatures [5].

Regarding polystyrene (one of the most used polymers worldwide), its molecular weight decreases at temperatures above 250 °C and over 300 °C volatile products are formed [6,7]; its thermal pyrolysis mechanism has not yet been completely understood but is considered to be a chain radical process that comprises the well-known free radical steps of initiation, propagation, and termination (see kinetic scheme in the Section Kinetic Mechanism in the Appendix A). In the initiation step, radicals are generated by a diversity of mechanisms like chain-end scission, random scission, and ruptures at the called “weak links” that exist in the polymer structure such as head to head linkages, branches, and unsaturated bonds which are formed during polymerization, aside from the normal head to tail propagation process [7,8,9,10]. Propagation reactions that take place during PS pyrolysis are multiple: intramolecular and intermolecular transfer of hydrogen atoms, de-propagation, and β-scission; among these, β-scission is probably the most frequent in the range of 280 °C to 350 °C, taking place either at the end of the polymer chains or at a mid-chain position with a characteristic reversible nature, its rate increases as the ceiling temperature (T_c_) is approached. PS ceiling temperature has been reported to be 310 °C [11], 277 °C in gaseous conditions, and 397 °C when melted [12] (the T_c_ is the temperature at which propagation and depolymerization rates are equal). Hydrogen abstraction is the second most important type of reaction in the thermal PS pyrolysis; when a hydrogen atom is abstracted from the main chain by any radical, the attacking radical gets a saturated chain and a new radical is formed which later undergoes β-scission [12]. Intramolecular hydrogen transfer reactions are predominant at pyrolysis temperatures between 600 °C and 700 °C, promoting the production of benzene, toluene, ethylbenzene, and α-methyl styrene [13]. Another important reaction is called de-propagation, also known as unzipping or depolymerization, in which polymer molecules break down unit by unit by a chain radical mechanism, it is known to produce lower polymers and is the main mechanism responsible for the production of styrene monomer [7,14]. Termination reactions have been reported to include recombination [15,16] and disproportionation [17], although their quantification at pyrolysis temperatures is uncertain.

From published studies on the thermal degradation and pyrolysis of PS where a spectrum of molecular weights in a variety of reactions systems and set of conditions have been evaluated [9,18,19,20,21,22,23], it has been concluded that the main products obtained are: benzene, toluene, ethylbenzene, styrene, cumene, and diphenyl propane, among other aromatic components whose abundance is reflected as a function of temperature [19]. Regarding styrene monomer production, its generation will be favored when low pyrolysis temperatures are used; at high ones, its production will be notably reduced, increasing at the same time the amounts of produced toluene and ethylbenzene [24].

One hypothesis that has already been proved right by our previous studies [1] is that the introduction of a nitroxide moiety at the end of the polystyrene chains increases the probability of initiating depolymerization reactions on these labile structures when subjected to a heat treatment, more specifically at the oxygen-carbon bond between the nitroxide and the polymeric chain-end at temperatures that are slightly higher than the ceiling temperature to promote the depropagation reaction, maximizing the production of styrene monomer. This occurs in addition to the initial rupture at other parts of the structure that also leads to depropagation in these conditions.

The technique employed to attach the nitroxide species to the end of the PS chains is the Nitroxide Mediated Polymerization (NMP), which enables the production of polymers mainly constituted by a nitroxide end-functionalized dormant polymer with defined molecular weight and narrow polydispersity. Its main feature is a reversible termination mechanism that takes place between propagating growing species (polymer radicals) and a stable free radical (nitroxide radical) that controls the polymerization by generating alkoxyamines (dormant species) as dominant species. The dormant species are then reactivated generating propagating radicals and nitroxide radicals through a homolytic breakage as the dormant and the radical species are in dynamic equilibrium resulting from the reversible deactivation-activation reactions. Studies on the pyrolysis of polystyrene synthesized with benzoyl peroxide (BPO) and TEMPO (2,2,26,6 tetramethylpiperidin-1-yl)oxyl) as nitroxide reveal that its degradation occurs at 400 °C, very similar to the degradation of non-nitroxide polystyrene, but with the noticeable difference that a mass loss is observed below 300 °C, suggesting the rupture of the PS-Nitroxide bond and the breakage of the N–O bond of the TEMPO moiety [25]. The lack of knowledge of the position and nature of the initial scission in polystyrene thermal degradation has restricted its quantitative analysis.

### 1.2. Background: Mathematical Modeling of the Thermal Pyrolysis

The development of a mathematical model of a reaction and process provides us with a handy tool that captures the essence of a system that allows the understanding of the involved variables, and at some level, the prediction of the reaction taking place. The pyrolysis of PS is a complex problem, and although various models have been proposed through the years, no definite mechanism nor model has been established. Among these, it is worth mentioning some interesting studies: McCoy and Madras [26] models are based on random-chain, mid-chain and end-chain scission mechanisms, and they solve the MWD as a function of time from a batch reactor population balance equation. Their results illustrate how the MWDs decrease in time while the dispersity increases due to the formation of smaller molecular weight products. Westerhout and coworkers [27] examined the degradation kinetics of polyethene, polyisoprene and PS, and they also evaluated two mathematical models with the aim to determine the kinetic data for these materials pyrolysis: a first-order model and a random chain dissociation mode. The first one, a first-order power-law, was used to enable comparison with the literature data and was found to be applicable only in a small conversion range, the second model incorporates a statistical reaction pathway model in which different types of bonds are distinguished, each bond having different breakage rates with different kinetic parameters. The most important difference between these two is that the latter accounts for the presence of weaker bonds in the polymer chain by considering side chains and different types of bonds and can be used to predict the product spectrum of the primary pyrolysis reaction. Sterling and coworkers [28] developed the continuous distribution kinetics technique for experimentally determining polymer degradation reaction mechanisms and rate parameters; the polystyrene thermolysis was studied through moderate conversion where a random scission rate coefficient was assumed not independent of the molecular weight. Kruse, Wong, and Broadbelt [29] developed population balance equations and used the method of moments to model the degradation of polystyrene at a mechanistic level; full approximated MWD’s were constructed from the zeroth, first, and second moments tracked for each species within the PS model using Schultz and Wesslau distributions. The reactions taken into account included chain fission, hydrogen abstraction, mid-chain β-scission, end-chain β-scission, 1,5-hydrogen transfer, 1,3-hydrogen transfer, radical addition, bond fission, radical recombination, and disproportionation. Meanwhile, polymer species were lumped into various classes to track the presence and location of radical centers, the position of double bonds, the inclusion of branches, and the orientation of the “head” and “tail” ends of the monomer units. The program was constructed as a set of ordinary differential equations that included three moment equations for each unique species. The full model included over 2700 equations. Song [30] conducted an optimization study on the PS pyrolysis in solution in a batch reactor where the aim was to find the optimal temperature profiles minimizing the reaction time and the process energy for the desired conversion; the PS pyrolysis was represented by the binary scission at any position along the chain and chain-end scission with the release of monomeric species; to represent the MWDs of the polymers and products a gamma distribution function was used. The continuous kinetic models included the assumption that the degradation reactions were irreversible of first order, and that all of them had equal reactivity irrespective of the chain length of the polymer molecules.

As McCoy stated [26], pyrolysis can be seen as a fragmentation phenomenon and monitoring the evolution of the MWD and its moments in time provides considerable information and allows a sharper interpretation of the kinetics and mechanism of the degradation reactions.

In our study, we aim to acknowledge the time evolution of the full polymer chain size distribution generated by the thermal decomposition of PS using the kinetic model proposed in our previous studies, which takes into account the following reactions: mid-chain random scission of dead polymers, transfer to polymer and β-scission, de-propagation, end-chain scission for dead polymer, termination by combination, and termination by disproportionation. It also includes the reactions that involve the nitroxide species: activation-deactivation of dormant polymer and mid-chain random scission for dormant polymer. To this end, a mathematical model of the PS pyrolysis was developed as a set of differential equations, whose solution was implemented in FORTRAN adjusting the kinetic parameters to fit some of the responses experimentally measured. The PS pyrolysis experiments were performed in a system similar to the one used in our previous work, where a precise set of conditions and temperature ramps were defined to maximize the production of styrene monomer. The pyrolyzed polymers were first synthesized by Free Radical Polymerization (FRP) and by Nitroxide Mediated Polymerization (NMP).

## 2. Materials and Methods

### 2.1. Materials

Styrene (99%), benzoyl peroxide (BPO) (98%), (2,2,6,6-tetramethyl-1-piperidinyl-1-yl)oxyl (TEMPO) (98%), industrial grade methanol, toluene and acetone, tetrahydrofuran (inhibitor free, suitable for HPLC, >99.9%). All reagents and solvents by Sigma-Aldrich, St. Louis, MO, USA).

### 2.2. Experimental Methods

#### 2.2.1. Polystyrene Polymerization

Polystyrene samples were synthesized in two different ways, by the Nitroxide Mediated Polymerization (NMP) (at three different molar ratios of nitroxide to initiator, N/I = 0, 0.9, 1.1, 1.3), and the Free Radical Polymerization (FRP) techniques, each one used different reaction temperatures and took contrasting polymerization times; yet, both aimed to obtain a molecular weight average in number (M_n_) of 50,000 Da. All polymerizations were conducted in bulk using a batch process. The process and conditions were identical to those described in a previous publication of our group [1] (see Appendix A for more details).

#### 2.2.2. Polystyrene Pyrolysis

The used depolymerization reaction system, shown in Figure 1, is almost identical to the one used in our previous studies [1], with the important difference that all the valves and connections of the reactor vessel are welded (or perfectly Teflon^TM^-sealed), making a complete unit and preventing the produced reaction fumes from escaping the system from other places than from the condenser. The system essentially consists of a 50 mL stainless steel reactor vessel that is heated homogeneously by an electric mantle. The complete description of the system, the details of the temperature control and the peripheral equipment, as well as the detailed pyrolysis procedure are provided in the Appendix A; here, only a summary is given.

The pyrolysis experiments were performed in two stages, first, a set of exploratory pyrolysis reactions were conducted to define a set of reaction conditions and a process that involves fixed temperature ramps that help to maximize the amount of monomer recovery on the product liquid fraction. Then, using the previously established temperature ramps and set of conditions, pyrolysis reactions were performed using the synthesized polystyrenes.

The procedure for the pyrolysis of PS that was refined from the exploratory experiments is very similar to that reported before by us with one important difference. The process can be summarized as follows: 10 g of polymer powder were placed in the reaction vessel and nitrogen gas was passed through the system to remove oxygen from inside the reactor, then all system valves were closed and sequential well-defined temperature ramps were applied through the heating mantle to reach the desired pyrolysis temperatures. The key important difference with respect to the process previously reported by us lies in the heating ramps used in this case in which the heating procedure was implemented as follows (see a summary in Table 1):

The temperature controller was set in manual mode and its set-point was initially set at 50 °C. When the temperature of the reactor reached 35 °C (starting from ambient temperature) the set-point was set at 100 °C and when 85 °C was reached a third set-point at 150 °C was programmed. Given the thermal inertia of the system, the temperature surpassed the 150 °C of the set-point and reached a value in the range 300–330 °C without further heating. The temperature control system consisted only of a heating mantle, so when the set-point was reached, the heating stopped, but there was not a cooling device to avoid further increase of the temperature. Once the temperature of the system stabilized around 300–330 °C, the set-point was set at 345 °C and the heat provided in this stage was sufficient to reach 390 °C for the pyrolysis reactions designed to be run at this temperature. During most of the heating stage, the valve between the reactor and the condenser remained closed. Only when the pressure of the system reached 12 psig and the temperature was between 330 and 350 °C was the valve open to allow the flow of vapors to the condenser.

Once the final reaction temperature is reached, it is maintained constant by setting the (on-off) controller in automatic mode at the desired set-point until all the material pyrolyzes. The reaction was considered finished once no more liquid material was recovered from the condenser. It is important to point out that if heating ramps with ΔT higher than 15 °C are used from 320 °C on, recombination of light components will predominate, diminishing the amount of monomer present in the liquid phase. Even though solid, liquid, and gas products were produced during the pyrolysis reaction, only the liquid phase was recovered at the end of the reaction and characterized, while the solids were only weighed.

The liquid phase obtained from the pyrolysis was analyzed and characterized by gas chromatography-mass spectrometry in an Agilent Technologies equipment Santa Clara CA, USA, model 7890 GC/MSD 5977B fitted with an HP-5MS-30m column of 0.25 mm internal diameter, 0.25 µm of pore size (Agilent Technologies), and calibrated with a PFTBA standard. The sample injection volume was 1.5 mL with helium as the carrier gas with a flux of 1 mL/minute. The injection temperature was 250 °C and the sample running temperature was from 80 to 300 °C (10 °C/min).

### 2.3. Modeling and Simulation Methodology

In this study, a kinetic model previously proposed by our group is used [1]. The model is based on the different chain-rupture mechanisms that take place on polystyrene when submitted to a thermal process and initiate the pyrolysis by creating free-radicals, and the different reactions arising from the interactions between different species such as dead polymer (D), living polymer (P), dormant polymer (S), living polymer with a nitroxide at the end (R), styrene monomer (M), monomeric radicals (M*), and nitroxide radicals (N).

Many different reactions arise from the application of a mechanism to different combinations of interacting species; such mechanisms include scission, chain-transfer, and termination reactions. As an example, if a dormant polymer (that comes from nitroxide end-functionalized polystyrene) undergoes scission at a middle-chain position, then two polymer chains are formed, one of them is a living one and the other is a living chain with a terminal nitroxide, both of them with active centers (active free radicals). These resulting chains may depropagate (generating styrene monomer), recombine with other species that possess active centers, or even react with chains that do not possess active centers through chain transfer (as in dead or dormant species), generating then, other sets of species like dead, dormant, living with nitroxide, monomer or radical ones. The possible mechanism combinations that can take place and aim to explain the decomposition of both of the synthesized PS samples in this work, via FRP and NMP, can be found in our previous study [1] and included for convenience in the Appendix A.

From these mechanisms, kinetic balances that represent the variation of each one of the involved species on the rupture and recombination mechanisms of the PS pyrolysis are developed resulting in a large set of ordinary differential equations (ODE’s). The numerical solution of this system of equations provides the time evolution of the small species and chain length (*n*) distributions for the different polymeric species generated through the PS thermal degradation. The equations were solved with a code written in FORTRAN using the routine DDASSL (Differential-Algebraic System Solver) [31] which is a Gear type method based on backward differentiation formulas of variable step–variable order for solving systems of differential-algebraic equations.

### 2.4. Kinetic and Mathematical Model

The equations of the mathematical model used in this work are very similar to those previously published by us [1], but there are significant improvements over the original model that should lead to a better mathematical representation of the systems under consideration. The resulting model includes differential equations for dead polymer (*n*), dormant polymer (*n*), living polymer (*n*), living polymer with nitroxide (*n*), monomer, monomeric radicals, nitroxide radicals and ethylbenzene, which are included in the Appendix, where (*n*) after a species indicates that they are chain-length (*n*) dependent distributions. The differences of the present equations with respect to the previous ones are the following:


(1)In scission terms, the probability of scission at a specific position along the chain is assumed to be uniform (equally likely at any position); therefore a factor of the inverse of the number of possible scission points should affect the corresponding term. In our previous modeling, as an approximation, this number was assumed to be *n*, that is, the number of repeating units in a given length-*n* polymer chain because this facilitated the math to derive the moment equations, so the factor used was 1/*n*; however, the scission occurs at a link (bond) *between* repeating units, and this number is only *n* − 1, therefore, the correct factor should be 1/(*n* − 1). The use of the approximated factor 1/*n* should not introduce a significant error for long chains which is the case at the beginning of the pyrolysis process; nonetheless, as the pyrolysis proceeds and the chains become shorter, the error is magnified and can become quite significant. In the improved model, this error is eliminated by using the correct 1/(*n* − 1) factor. Although not used in the calculations for the present work, the corrected moment equations were re-derived and they are included in the Appendix.(2)Another important difference with respect to our previous treatment of these equations is that in the previous work only the moments of the distribution were solved using some approximations to deal with the closure problem of the moments, while in this work the equations are solved for the full distributions without assumptions, therefore providing more accurate results. In this implementation, the moments, where needed, are calculated by direct application of their summation definitions (see Appendix B).


As input to the simulation program, the experimental MWD of the polymer to pyrolyze, as well as estimated kinetic parameters were fed. To represent the experimental MWD of the polymer fed (initial conditions for the equations), the distribution of dead or dormant polymer, depending on the type of polymer pyrolyzed, was approximated by using log-normal distribution functions.

Because pyrolysis simulations require solving a massive number of equations simultaneously (roughly four times the number of maximum length, *n_max*), the time necessary to complete each polymer simulation can be very long. As a consequence, and to verify the code and obtain a first kinetic parameter fitting to adequate the model simulation times to the experimental data, the model and simulations were first applied to the (hypothetical) pyrolysis of a styrene oligomer having an *M_n_* of 2000 g/mol. Once the model parameters were tuned for this system, the methodology was applied to simulate the pyrolysis of the real polymers used in the experimentations.

## 3. Results and Discussion

### 3.1. Experimental Results: Polymerization Product Analysis

The molecular weights, dispersities, reaction times, and conversions obtained from the polymerization reactions are summarized in Appendix A. The molecular weights of the synthesized polymer were in the range 37,000–48,100 (*M_n_*). Differences between the (planned) theoretical and experimental molecular weights are attributed to the initiator efficiency, mixing conditions, and reaction times. On the other hand, the MWD dispersities obtained from the experiments are a clear sign of good control in the case of NMP reactions (1.25–1.34) as they are expected to be between 1.1 and 1.3 [32] and in the non-nitroxide PS, the temperature and stirring were optimum as the dispersity was below 2 (1.75).

### 3.2. Experimental Results: Pyrolysis Product Analysis

Relative areas from chromatrographs were used to quantify the total amount of styrene and some other valuable molecules recovered after pyrolysis (styrene characterized by its mass spec). Typical results from previously reported processes are those shown on the TIC (Total Ion Chromatography) of Figure 2a revealing the presence of many impurities in addition to the styrene peak. Using our new reported methodology (NMP and FRP polystyrenes pyrolyzed at 390 and 420 °C), it can be clearly appreciated a styrene-rich output with just a few impurities (Figure 2b) after analysis at the set and above described chromatographic conditions. The conditions for the experiments performed in the exploratory and designed phases are described in Table 2.

The exploratory experiments were initially performed to reproduce and compare with our previous studies [1] to define if the slight process differences would provoke significant changes in the results. They correspond to the pyrolysis of one polymer sample (NMP, N/I = 0.9) at two different temperatures (390 and 420 °C). Initially, the composition results did not replicate the original ones with large differences in the yield of styrene; in the initial experiments, the styrene yield was around 38–40%, while in the new experiments the yield was only ~25%. This prompted a study to know the effect of the different variables involved in the process when they were varied one at a time. The variables that in the previous study had shown the most significant effects in the product composition were pressure, reaction times, and the time of release of light volatile products, but the variation of these parameters in the present study did not show any significant improvement in the styrene yield. Finally, after a closer look at the reaction procedure, it was realized that the key variable to increase the yield was the pace of heating or how the heating ramps were implemented. In the exploratory experiments, the heating rate was faster, with six intermediate set-points before reaching the set-point corresponding to the final target reaction value (390–420 °C). This resulted in an aggressive heating period of 6–7 min that gave rise to many undesirable pyrolysis byproducts. When the heating was provided more gradually, by setting intermediate temperature set-points before reaching the final target reaction temperature as described in Section 2.2 and Table 1, the thermal inertia of the system was taken into account avoiding the sudden supply of excessive energy in short times, resulting in smoother heating periods of 12–13 min. This allowed us to increase the styrene yields to around or above 60%, while the rest of the products were mainly toluene, α-methyl styrene, styrene dimer, and triphenyl cyclohexane; at this point, it was considered that the technique for the polystyrene pyrolysis was defined. Representative chromatographs of the previously reported process and the present one are compared in Figure 2; it is evident that the present process generates a styrene-rich output with few impurities while the previous process exhibits many impurities in significant yields. Using this technique, the previously synthesized NMP and FRP polystyrenes were pyrolyzed at two different temperatures: 390 °C and 420 ° C, given our previous experience. Figure 3 shows the percentage in weight of the three types of products (solid, liquid and gas) relative to the fed sample and their comparison to equivalent experimental results from our previous work [1]; they all exhibit higher amounts of liquid fraction.

Figure 4 shows the composition of the liquid fraction expressed as % wt. of the total polymer pyrolyzed. The implemented new pyrolysis technique is reproducible (each experiment was run by triplicate) and significantly reduces the amounts of byproducts, generating higher amounts of styrene. From Figure 3 and Figure 4 it can be seen that, in general, in the samples with N/I ratios of 1.3 and 1.1, the amount of recovered styrene is higher, as well as the liquid fraction compared to the FRP polymer. On the contrary, when pyrolyzing the sample with N/I of 0.9, the amount of obtained styrene was slightly lower than that from the FRP sample. The best overall results at 390 °C were those with a N/I ratio of 1.3, reaching values of 75% in weight of recovered styrene. Meanwhile, at 420 °C, the best results were the ones with the N/I ratio of 1.1 (68% wt styrene). This shows that although the monomer recovery of the FRP samples is high, the presence of nitroxide improves the monomer recovery with lower reaction times, except when the polymer was synthesized using the N/I = 0.9 ratio. A summary of the results in tabular form can be found in Table 3 and Table 4.

### 3.3. Mathematical Modeling Results

As mentioned before, the simulations were executed in two stages; first, a simulation of the hypothetical pyrolysis of styrene oligomers was carried out, and secondly, the simulation of the pyrolysis of the polymers used in the experiments.

#### 3.3.1. Oligomer Modeling Results and Parameter Values Used

For the simulations of oligomers, two different kinds are considered: a free radical oligomer (FRO) and a nitroxide mediated oligomer (NMO); their initial molecular weight distributions were constructed assuming a Flory distribution with an average molecular weight *M_n_* = 2000 g/mol (~19 repeating units).

The Flory distribution can be expressed as in Equation (1) where $N_{n}$ is the number of polymer molecules with *n* monomeric units and *P* the probability of propagation that can be obtained by relating the number-average degree of polymerization (or *M_n_*) by means of Equation (2), where *M*_0_ is the molecular weight of the monomer.
(1)$$N_{n}=P^{\left(n-1\right)}\left(1-P\right).\;$$
(2)$$M_{n}=\frac{1}{\left(1-P\right)}M_{0}$$

Once the value of *P* is calculated, the molecular weight distribution generated for oligomers from Equation (1) is assigned as the initial condition for the dead chain oligomer species (D) for the FRO and as the dormant polymer chain (S) for the NMO, leaving the rest of the initial values of the distributions equal to zero: *P(n) = R(n) = S(n) =* 0 for FRO, and *D(n) = P(n) = R(n) =* 0 for NMO, for all *n =* 1, 2, 3.

Initially, the used kinetic constants to run simulations for both types of oligomers were the ones defined in our previous work in which the method of moments was used [1]. However, when these were tested in the current model (complete MWD’s using ODE’s) for both samples, they resulted in material degradation times lower than a second to be converted into monomers and monomeric radicals. A more thorough review of the values for the kinetic constants used previously led to the conclusion that some of them were not realistic and even created some numerical problems when used with the present model. In particular, the rate coefficients for end-chain and mid-chain scission used before seem to have been seriously overestimated by around four orders of magnitude, leading to extremely short chain-degradation times when used in the model. Notice that, as pointed out in our previous publication, the values of the kinetic parameters used in that work should have been taken with caution since they came essentially from an overall parameter sensitivity study; other combinations of parameters could have led to similar qualitative results. A similar sensitivity study was run with the present model and this led to a new set of kinetic parameters which are shown in Table 5 that allowed the reaction dynamics predicted by the model to fit better the reaction times experimentally observed. The only two criteria used to select the values of the kinetic parameters were that these values could correctly predict the overall degradation time and the overall monomer yield recovery within reasonable limits. The kinetic parameters were then fitted to allow overall degradation time of the oligomers and conversion to monomer and radicals about 20 and 24 min for NMO and FRO respectively. Given the limited measurements available, no attempt was made to fitting temperature-dependent Arrhenius type expressions for the rate coefficients. In the following section, the main findings of the parameter sensitivity study are summarized and briefly discussed. Additionally, the mid-chain scission rate constant was compared with that reported by Kruse et al. [33] and their value and ours are comparable at 420 °C (6.0 × 10^−6^ and 1.0 × 10^−5^ s^−1^ respectively), giving more confidence to our present estimation. It is important to mention that the rate constant values in [33] were theoretically estimated so they are not experimentally determined values. They used a frequency factor and an estimation of the activation energy based on the heat of reaction (according to Polanyi [34]), which was itself estimated using the heats of formation of the involved compounds calculated from public databases.

Regarding computation times for the oligomer pyrolysis, these were ~5 min for the free radical oligomer and ~45 min for the nitroxide mediated oligomer, confirming the advantage of using an oligomer model for parameter fitting since the computation of high polymer pyrolysis finally took 15–20 h in a laptop computer with an I7 Intel RM processor. The number of ODE’s solved was ~800 for the oligomer simulations and ~6000 for the high polymer with a maximum chain length of 1500 units, equivalent to chains with a molecular weight of ~156,000 Da, about thrice the *M_n_* of the starting material.

#### 3.3.2. Parameter Sensitivity

The parameters that exhibited the largest influence on the overall degradation time were $k_{b}$ and $k_{be}\;$(mid-chain and end-chain scission, respectively) in the pyrolysis of FRO and $k_{a}$ (end-chain scission at the C–O bond of the nitroxide functionality) in addition to the former two for NMO. It was assumed (and this was subsequently proved) that the simulated degradation times would not be much different for the oligomers and the polymers (the *a-posteriori* actual time differences were within 1–2%, confirming the validity of the assumption). The observed sensitivity to the aforementioned constants is mechanistically understandable since it points out the initial radical generation of the chain (by bond breakage) as being the limiting rate step of the chain degradation process. The degradation time was also sensitive, to a lesser extent, to $k_{trb}$ which is also a radical-center generation process. The single most influential parameter was $k_{be};\;$ the degradation time significantly changed upon small variations of this parameter. It is also notable that the selected values of these rate coefficients induce almost total conversion of species to styrene and monomeric radicals with essentially no production of dimers or trimers (see further discussion on this issue below). On the other hand, for dormant polymer (or NMO) $k_{a}$ and $k_{d}\;$ both showed influence on the degradation time and were jointly fitted, suggesting that it is really their ratio (equilibrium constant) that is key to determine the process overall rate. However, once the $k_{a}\;$ value entered a certain interval, its effect reached a plateau and further changes around the mid-value of the interval of ±2 orders of magnitude exhibited little influence on the response. The values of the termination rate coefficients seem to be rather low, but using higher values would conduct to lower conversions for a fixed reaction time or longer reaction times for reaching a given conversion, not corresponding to the experimental observations. This kind of behavior was also present and discussed in our previous work [1]. It is also interesting to mention that Kruse et al. [33] report the use of termination constants affected by the gel effect, which is known to cause the reduction of the termination constant by several orders of magnitude.

Note. Since no detailed comparisons with experimental data were available, the rate coefficients should be roughly applicable in the temperature range 390–420 °C. No attempt was made of fitting Arrhenius-type expressions for the rate coefficients.

For the free radical oligomer pyrolysis, the dynamics of species conversion (Figure 5) is obtained at times similar to the experimental ones, highlighting that the reduction of dead chains (D1, first moment) is gradual and inversely proportional to the monomer (MS) and radical (MR) generation. Similar results were obtained for the NMO simulation (Figure 5) where the participation of the kinetic constants $k_{a}\;$ and $k_{d}$ have the effect of reducing the reaction times to 20 min compared to the 24 min of the FRO (like in the polymer experimental reactions), achieving almost total conversion into monomer and radicals, as expected. For a simpler graphical representation, the MWD data are converted to moments of the distributions for the different polymer populations (only moment 1 values are plotted). The moments are calculated from their definitions by taking summations on the calculated MWD results and those corresponding to the zero-th, first, and second moment of live polymer are denoted as P0, P1, and P2, respectively. Similarly, D0, D1, and D2 are used for the moments of dead polymer, while S0, S1, and S2 are for dormant polymer and R0, R1, and R2 for the live polymer with nitroxide. As explained before, the temperature dependence of the rate constants was not explicitly accounted for, but the reference experimental data (degradation times) were taken as those at 390 °C.

The full molecular weight distributions generated for both free radical and nitroxide oligomers do not show any multimodalities. For FRO, the distributions calculated for dead and living oligomer species (Figure 6) show the expected trend of chain-length reduction as the reaction time progresses, observing that the chain length that predominates for longer reaction times is of approximately three monomeric units after about 50 s of reaction, which shows that at relatively long reaction times the changes in the oligomer populations are gradual.

Similarly, the evolution of the polymer size distributions for the NMO pyrolysis (Figure 7) shows that most of the chain-length reduction occurs at relatively short reaction times; from 50 s on, the curves barely show any significant differences although the monomer generation evolves more gradually, which suggests that the components inside the reaction will have a chain length similar to two units most of the time and later they may break to generate styrene monomer. The rapid displacement of the curves to shorter chain lengths in all cases may indicate that chain scissions could take place almost instantaneously to smaller fractions in the first seconds of the reaction, and that these smaller chains will continue to more gradually break to produce monomer during all the reaction.

#### 3.3.3. Polymer Modeling Results

Once the values for the set of kinetic constants were defined in the previous section, polymer simulations were performed. To be able to replicate the characteristics of the synthesized polymers, molecular weight distribution curves were constructed based on the experimental MWDs and used as initial conditions for the simulations; these distributions were fitted to emulate the ones obtained by GPC experimentally and were generated using a log-normal distribution, which allowed replication of all the samples with good precision (see the fit of the calculated and experimental distributions in Appendix A).

#### 3.3.4. Comparison with Previous Simulations

To assess the quality of our previous calculations [1] comparisons between the previous model predictions and the current one were made by contrasting the evolution of the number average molecular weight (*M_n_*) and MW dispersity with reaction time for some representative cases (PS synthesized via FRP and NMP), (see Figure 8), where the effect of the changes and corrections implemented in the current model can be appreciated. The plotted *M_n_* value represents that of dead polymer for FRP-PS and the summation of dormant and dead polymer distributions for NMP-PS. The *M_n_* behavior is only qualitatively approximated by the previous model, but there are significant deviations from the exact behavior (represented by the new model) even at relatively short reaction times. From the insets in Figure 8, it can be seen that the degradation dynamics are much faster with the current model than with the previous one; the current model predicts initial degradations to oligomers in the order of fractions of a second to 1–2 s (faster for NMP-PS); while the previous model predicts the initial degradation to occur in the order of ~100 s or more. The deviations are more evident for the MWD dispersity which is grossly misrepresented by the previous model. These deviations are more evident at reaction times larger than 700 or 200 s for the FR and NMP cases respectively. They are attributed to the increasing error arising from the 1/*n* factor used in the previous model in the scission terms instead of the correct 1/(*n* − 1) factor, an error that, as explained before, becomes more important as the chains shorten at longer reaction times. These results indicate that the assumptions used in the previous model, especially those associated with the 1/(*n* − 1) factor, are not acceptable in this kind of process.

#### 3.3.5. Simulation of FRP Polymer Degradation

The simulations of the pyrolysis of FRP polystyrene showed dynamics in which all of the dead chain population degrades and turns into monomers and monomeric radicals almost in the same proportion (Figure 9a). Although the corresponding mechanism is not explicitly included in the model, it is conceivable that a significant fraction of the monomeric radicals could become dimer upon cooling of the reaction mixture via termination by combination. Meanwhile, the dead polymer distribution (Figure 9b and Figure 10a,b) shows a rapid decrease in polymer chain length, exhibiting a peak of the curve at 440 monomeric units, corresponding to a molecular weight close to 46,000 g/mol (very close to the experimental value) at a nearly initial time (0.02 s), while at 2.4 s the curve peak corresponds to 30 units and progressively, from 50 s on, the peaks of the curves have average values of two units (the plotted curves are normalized and therefore their areas are no longer proportional to the corresponding amount of polymer).

Plotting the *M_n_* profile (Figure 9b) obtained by dividing the first moment (D1) by the zeroth moment (D0) of dead polymer from the simulation data, the curve shows a very fast decrease in the molecular weight of the polymer in the first seconds of the pyrolysis and then a very small, gradual decrease, once low molecular weights are reached in the remaining reaction time. This behavior has a direct relationship with the MWD’s of Figure 10, in which populations of long chains are present at the beginning and short chains at the end of the reaction. Qualitatively, the same decreasing behavior of molecular weights is also observed for the simulated NMP samples at all N/I levels, where the main differences are observed at the initial molecular weight of each sample.

The MWDs corresponding to the living polymer population (Figure 10c,d) show similar behavior to the dead chain distribution, with a slight displacement to the left for the samples in 0.1 and 0.5 s in the length range between 2000 and 3000 units, which means that when the living polymer chains that result from ruptures in the dead initial polymer are formed, some monomeric units are lost as expected.

#### 3.3.6. Simulation of NMP Polymer Degradation

Reaction dynamics and distributions of all NMP pyrolyzed nitroxide mediated polymer samples and distributions were simulated, showing a similar behavior between the three evaluated levels of the N/I ratio (1.3, 1.1, 0.9). The reaction dynamics for the N/I = 1.3 ratio (Figure 11) shows a very rapid conversion of dormant chains to dead ones, which degrade progressively and behave as in the free radical polymer system in a smaller reaction time. Moreover, the concentration of species of living polymer, living polymer with nitroxide at the end and nitroxide radicals have very small values as expected for radical species.

Chain distributions of dead (Figure 12i), dormant (Figure 12ii), and living (Figure 12iii) chains show curves with similar behavior in all cases, where the peak in the initial time corresponds to the initial molecular weight and a later fast shift of the curves to low molecular weights in comparison with the simulations of free radical polystyrene. As an illustration of this, at 0.5 s, a 30 unit-length peak is reached in the nitroxide mediated polymer while a similar peak is obtained at 2.4 s with the free radical polystyrene, demonstrating that the presence of activation and deactivation reactions related to the nitroxide moiety accelerates the reaction time. Figure 12iv shows MDW’s for living chains with nitroxide, but for initial times the peak is located at 368 monomeric units, which indicates that these species appear after the decomposition of the dormant ones.

Looking closely at Figure 11 more insight can be obtained into the mechanism of degradation of dormant polymer (NMP-PS) as opposed to that of dead polymer (FRP-PS). At a first look, it would seem that all the dormant polymer is quickly activated and converted to dead polymer via termination reactions and this would result in little or no advantage of having dormant polymer as starting material for the pyrolysis. However, looking at the inset in Figure 11, it turns out that all the dormant polymer instantaneously becomes live polymer and the sudden rise in the concentration of polymeric radicals induces fast termination reactions between pairs of these radicals to become dead before they are slowly reactivated by breakage reactions; however, some of the polymeric radicals undergo degradation (that competes with the termination reaction) and it is in this fraction of the population that some advantage results in the pyrolysis of dormant polymer in comparison with that of dead polymer as starting material, since some polymer is degraded faster during the period of high concentration of polymeric radicals; notice in the inset that the peak of the moment 1 for dead polymer (D1) is around 400 moles, while the initial condition for S1 is above 450 moles.

The time evolution of dead, dormant, living, and living with nitroxide-end species and their molecular weight distributions for the other NMP-PS prepared (N/I = 1.1 and 0.9) showed the same qualitative behavior as the N/I = 1.3 sample when simulated. Moreover, the conversion to monomer and radical monomers remained close to 97% of the initial polymer.

The kinetic parameters used in the simulation of the pyrolysis of polystyrenes prepared with the FRP and NMP techniques are able to replicate the high conversion to monomer and other small species observed in the experiments. Although the conversion to liquid products obtained experimentally oscillates around 85%, if the gaseous products are added (which should consist of small molecules), an average polymer conversion of 99% should be obtained, which reveals that the model predictions are consistent with the experimental results and represent a first approach to simulate the PS pyrolysis without approximations. Further work is needed to incorporate more detail in the mechanism and the model, especially to predict the yields of dimer and trimer species, although these changes should not represent a significant increase in the computational burden of solving the model. Regarding computation times, executing the simulation of the dead FRP polymer required around 15 h since these involved only two polymer populations and 15-18 h for nitroxide polymers that involved four polymer populations. In fact, for both cases, the same program was run with ~6000 ODE’s to solve (1500 equations per polymer population times four populations), but in the FRP case, all the rate coefficients related to nitroxide reactions were set to zero, alleviating somehow the computational burden. Moreover, once more level of detail is incorporated in the model for the chemistry of small species, the kinetic rate coefficients can be optimized using more rigorous methods for parameter estimation. One of the main goals of this work was to demonstrate the feasibility of solving the full MWD without simplifications with existing software and standard personal computers in reasonable computation times.

To our knowledge, this is the first time that the full MWD is calculated in an “exact” way (within the limits of numerical error tolerance) for the PS pyrolysis, in the sense that the full set of ODE’s representing the mass balances for species of different lengths was integrated without simplifications. As discussed in the introduction, in previous work, Kruse et al. solved the balances using the method of moments and then reconstructed an approximation to the full MWD by fitting Schultz and Wesslau distributions to match the first three moments of the distribution. This last approach introduces two sources of error: the third-moment expression used to solve the moment closure problem and the approximation of the distribution to a closed-form expression. In the present approach, an approximation-free solution is generated at a computational cost that is affordable with present-day standard computers. On the other hand, the model used by Kruse et al. is more detailed than the present one since it includes 1,3 and 1,5 hydrogen transfer reactions which, followed by β-scission, are responsible for the formation of dimer and trimer species respectively [33,35]. However, the incorporation of these reactions should not have a significant impact on the computational burden of solving the equations and it will be done in future work of our group.

## 4. Conclusions

A process for the efficient thermal pyrolysis of PS at relatively low temperatures (390–420 °C) has been developed and demonstrated with model polystyrene synthesized by FRP and NMP. The new process significantly improves the yield of styrene monomer, which is the most desirable target product in the pyrolysis process, compared with our previous process [1]. The styrene yields with respect to polymer fed increased from 28–39% in the previous process to 58–75% with the optimized process, and this was obtained by implementing an optimized heating ramp less aggressive than the ramp employed in our previous process. This improved process is highly competitive and simpler than other existing processes reported in the scientific and patents literature. It was confirmed that the presence of nitroxide groups in the PS provides clear advantages during the pyrolysis process in comparison with that of conventional PS synthesized by a free radical process, improving the yield of styrene recovered and reducing the overall reaction time. The best process results were observed when pyrolyzing NMP-PS at 390 °C using an N/I ratio of 1.3, reaching an average monomer recovery of 75% relative to the fed polymer sample, and reducing the processing time by almost 20% in comparison to the polymer synthesized via FRP. Regarding the modeling and simulation of the process, an improved model was presented and solved to generate for the first time the evolution of the full molecular weight distribution without any approximation by direct integration of a large set of ODEs representing the concentrations of polymeric species during the pyrolysis process. Initial parameter estimation was done after a parameter sensitivity study and the estimates of the kinetic parameters allowed to simulate reaction times and monomer yields similar to those experimentally observed. The simulation results indicate that the activation of the polymer that leads to further chain degradation by different mechanisms occurs in very short times, inducing a rapid decrease of the polymer chain-length until they reach values of average degree of polymerization equivalent to dimers and trimers, which then progressively break to form monomeric units during the reaction. Further refinement of the kinetic mechanism and the model is required to obtain more detailed predictions of small species, in particular dimers and trimers, but these changes should not represent a significant difficulty for the model solution and are already being implemented in work undergoing in our lab.

## Figures and Tables

**Figure 1 polymers-14-00160-f001:**
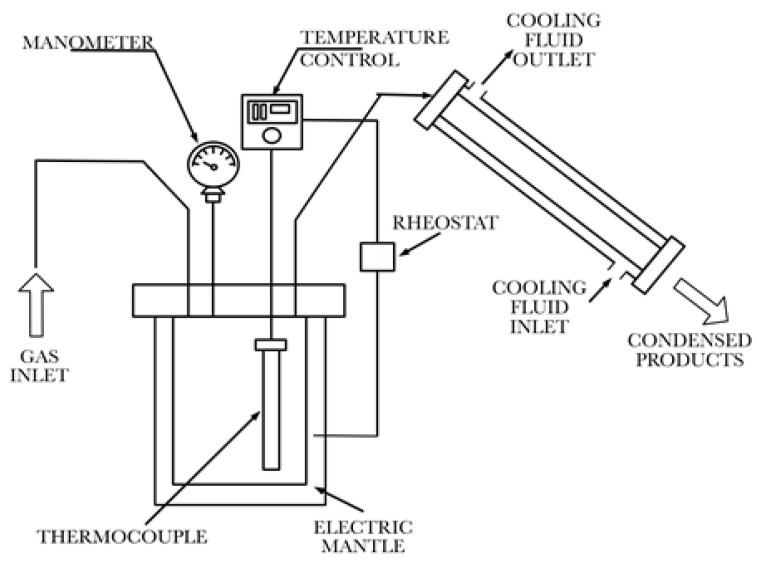
Pyrolysis reaction system.

**Figure 2 polymers-14-00160-f002:**
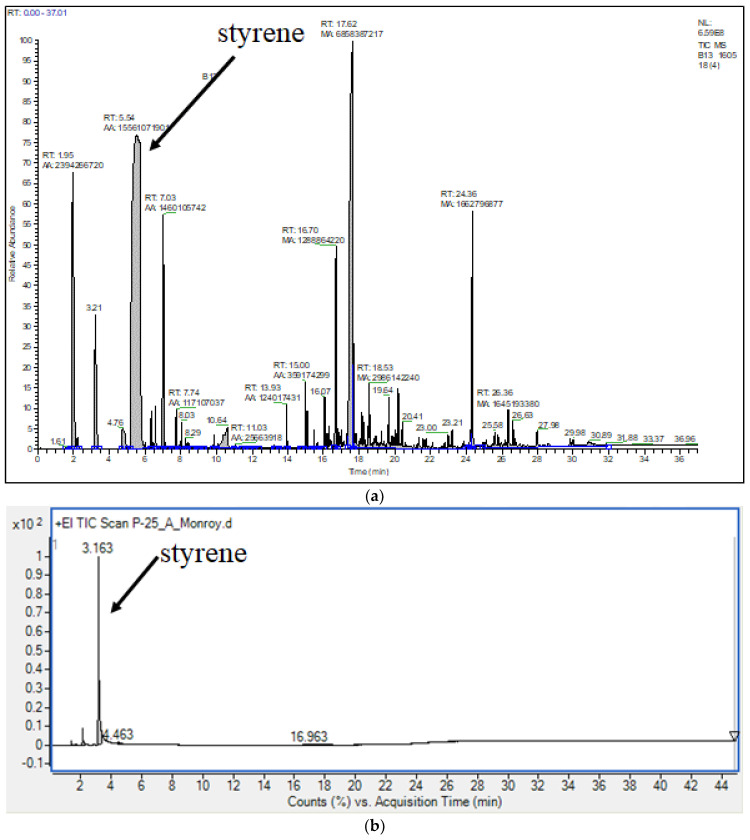
Gas chromatographs of the liquid fraction pyrolysis product: (**a**) previous process; (**b**) current process.

**Figure 3 polymers-14-00160-f003:**
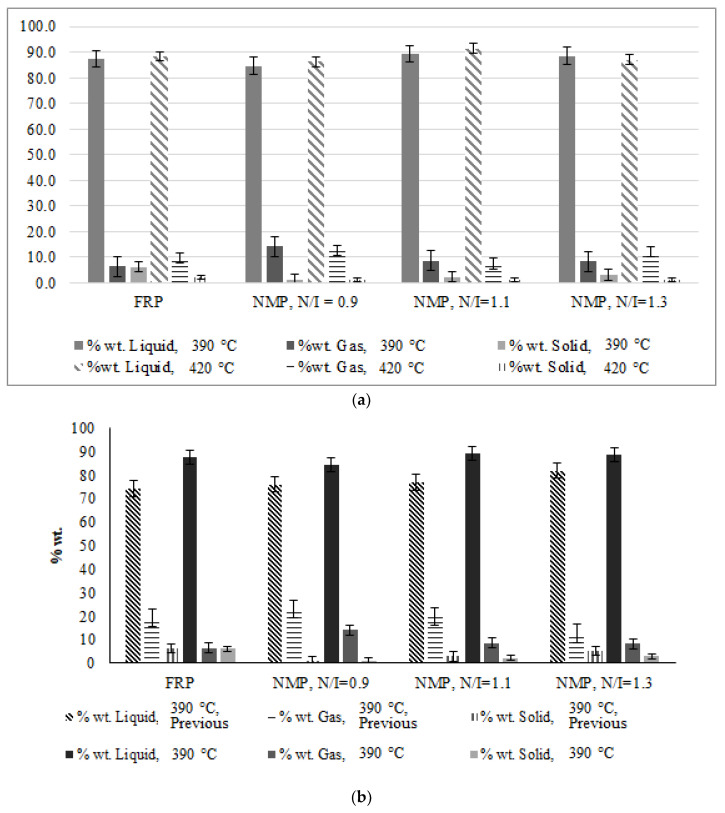
(**a**) Recovered pyrolysis products in the form of liquid, gas, and solids in %wt. relative to the fed sample at 390 °C and 420 °C, (**b**) Recovered pyrolysis products in the form of liquid, gas, and solids in %wt. at 390 °C and their comparison with previous experiments, (**c**) Recovered pyrolysis products in the form of liquid, gas, and solids in %wt. at 420 °C and their comparison with previous experiments. The experimental conditions for all cases are given in Table 2.

**Figure 4 polymers-14-00160-f004:**
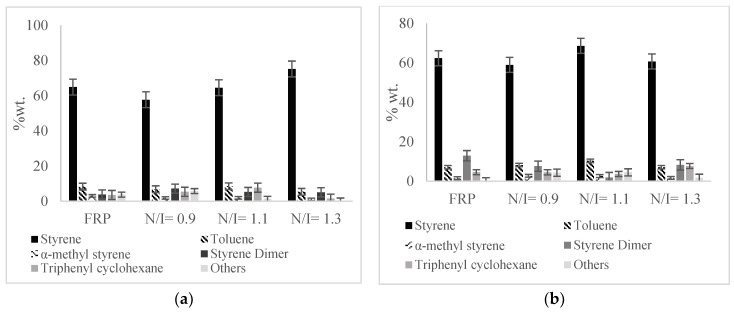
Composition of the liquid fraction in % wt. relative to the fed polymer at different pyrolysis temperatures: (**a**) 390 °C, (**b**) 420 °C.

**Figure 5 polymers-14-00160-f005:**
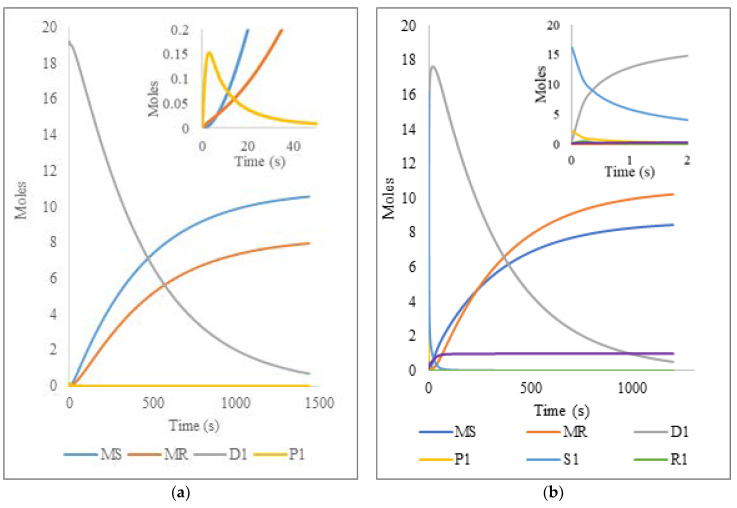
Variation of the species MS (monomer styrene), MR (monomeric radical), D1 (first moment, dead oligomer), P1 (first moment, living oligomer), S1 (first moment, dormant oligomer), R (first moment, living oligomer with nitroxide) and NX (nitroxide radicals) in the pyrolysis of (**a**) Free Radical Oligomer and (**b**) Nitroxide Mediated Oligomer.

**Figure 6 polymers-14-00160-f006:**
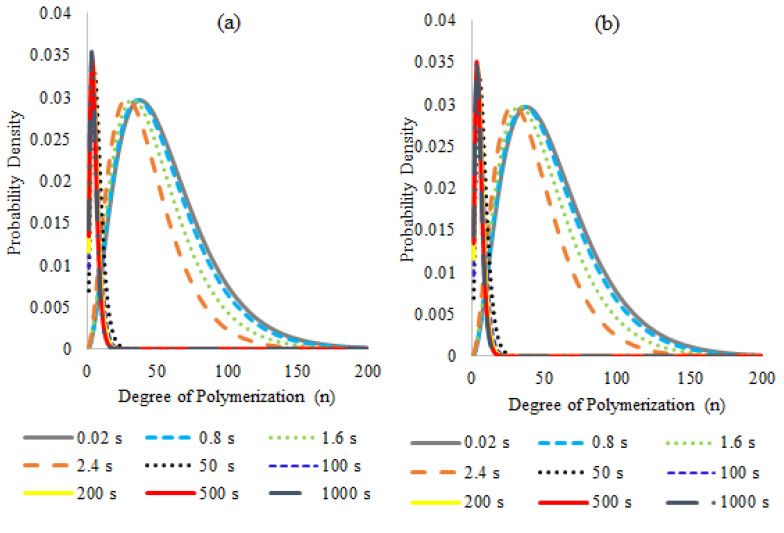
Molecular Weight Distributions calculated for (**a**) Dead chains and (**b**) Living chains of the Free Radical Oligomer pyrolysis simulation. Reaction temperature 390 °C (see footnote of Table 5) and reaction formulation as in Table 2.

**Figure 7 polymers-14-00160-f007:**
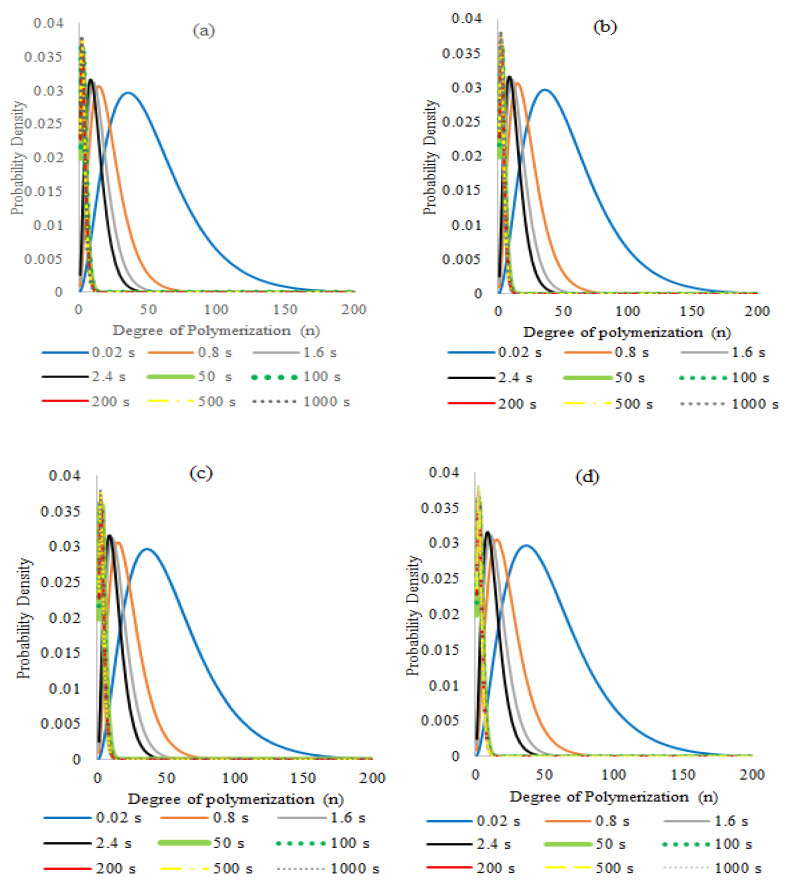
Molecular Weight Distributions calculated for the different polymer populations on the Nitroxide Mediated Oligomer pyrolysis simulation: (**a**) dead, (**b**) dormant, (**c**) living, and (**d**) living with nitroxide at the chain-end polymers. Reaction temperature 390 °C (see footnote of Table 5) and reaction formulation as in Table 2.

**Figure 8 polymers-14-00160-f008:**
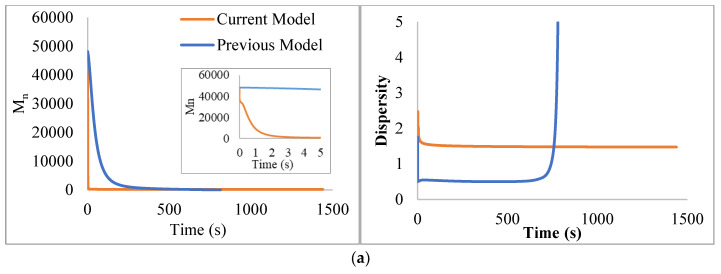
Comparison of M_n_ and dispersity evolution in reaction time of the pyrolysis simulation of polystyrene: (**a**) synthesized via FRP, and (**b**) synthesized via NMP with N/I = 1.1. The temperature for the calculations was assumed to be 350 °C.

**Figure 9 polymers-14-00160-f009:**
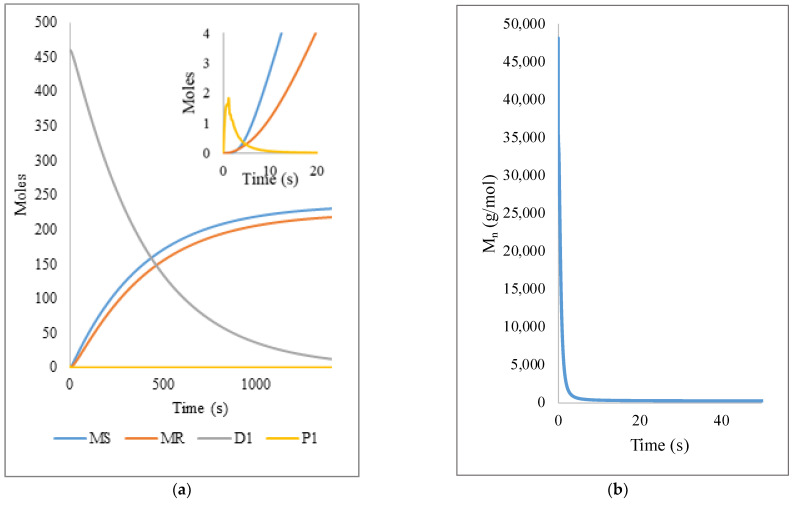
(**a**) Variation of MS (styrene monomer), MR (monomeric radical), D1 (dead polymer), and P1 (living polymer) in the pyrolysis of FRP-PS. (**b**) Time evolution of the number average molecular weight of the polymer in the pyrolysis of FRP-PS. Model predictions at 390 °C and reaction formulation as in Table 2.

**Figure 10 polymers-14-00160-f010:**
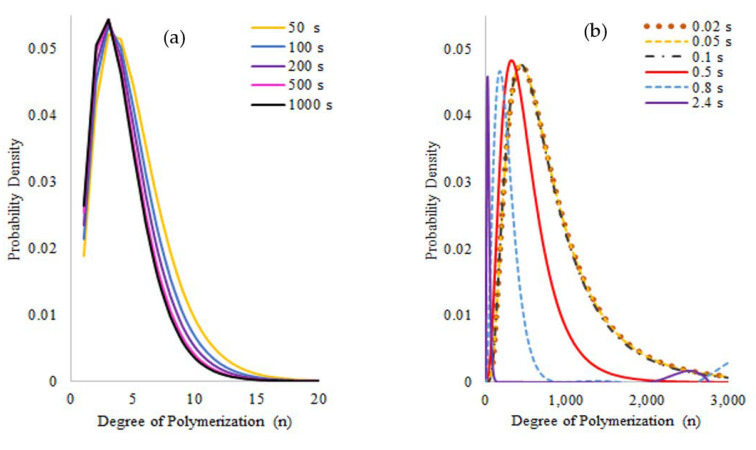
Molecular Weight Distribution of dead (**a**,**b**) and living (**c**,**d**) chains in the simulation of the pyrolysis of FRP-PS. Model predictions at 390 °C and reaction formulation as in Table 2.

**Figure 11 polymers-14-00160-f011:**
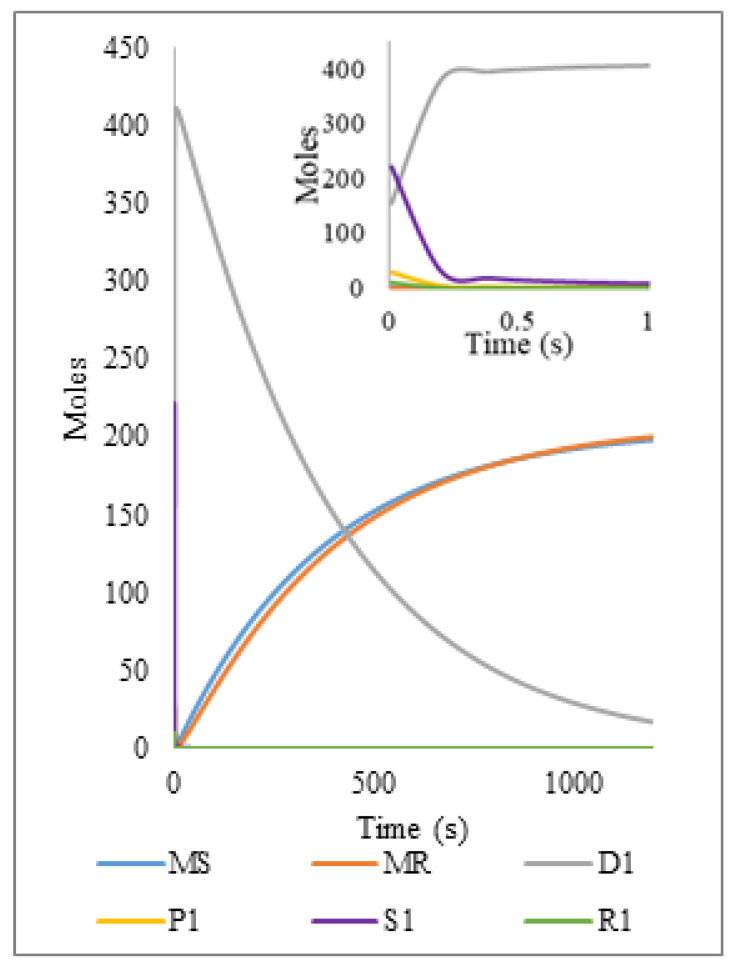
Simulation of reaction dynamics for the pyrolysis of NMP-PS (N/I = 1.3); time variation of MS (styrene monomer), MR (monomeric radical), D1 (first moment of dead polymer), S1(first moment of dormant polymer), R1(first moment of living polymer with nitroxide) and NX (nitroxide radicals). Model predictions at 390 °C and reaction formulation as in Table 2.

**Figure 12 polymers-14-00160-f012:**
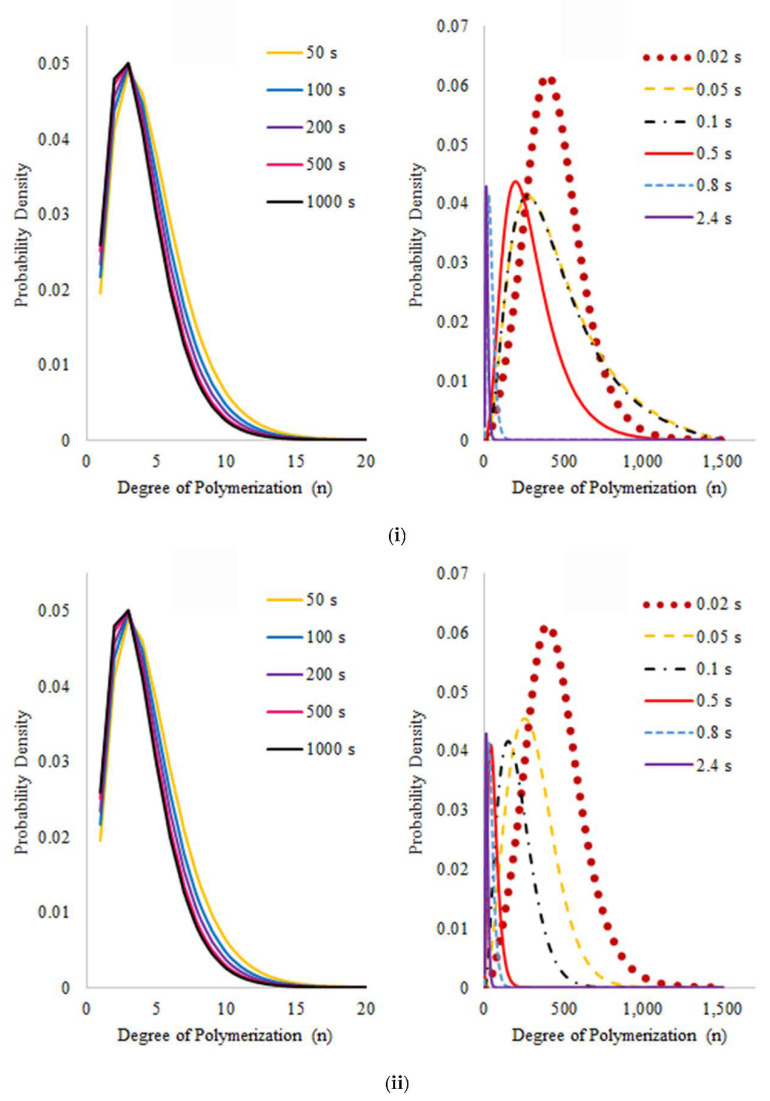
Molecular Weight Distributions of (**i**) dead chains (D), (**ii**) dormant chains (S), (**iii**) living chains (P), and (**iv**) living chains with nitroxide (D) in the simulation of the pyrolysis of PS synthesized via NMP, (N/I = 1.3).

**Table 1 polymers-14-00160-t001:** Summary of heating ramp strategy.

State of the System (temperature, °C)	Set-Point, °C
Ambient (18–22)	50
35	100
85	150
300–330	345
390	~400 (only for 420 °C reactions)

**Table 2 polymers-14-00160-t002:** Experimental design used in the pyrolysis experiments. All the experiments were run in triplicate.

Polymer	Pyrolysis Temperature, °C
FRP	390
NMP, N/I = 0.9	390
NMP, N/I = 1.1	390
NMP, N/I = 1.3	390
FRP	420
NMP, N/I = 0.9	420
NMP, N/I = 1.1	420
NMP, N/I = 1.3	420

**Table 3 polymers-14-00160-t003:** Yields of liquid, gas, and solids fractions in the pyrolysis reactions of FRP-PS and NMP-PS and reaction times.

T (°C)	N/I	% wt. Liquid	% wt. Gas	% wt. Solids	Time (min)
390	0	87.4	6.45	6.15	24
	0.9	84.6	14.2	1.2	19
	1.1	89.3	8.5	2.2	16
	1.3	88.6	8.3	3.1	19
420	0	88.4	9.5	2.1	17
	0.9	86.3	12.7	1	18
	1.1	91.5	7.5	1	17
	1.3	87.1	11.9	1	16

**Table 4 polymers-14-00160-t004:** Yields of styrene and other main components recovered in the liquid fraction in the pyrolysis reactions of FRP-PS and NMP-PS. The percentages are based on the total polymer fed.

T (°C)	N/I	% Styrene	% Toluene	% α-Methyl Styrene	% Dímer	% Tri Phenyl Cyclohexane	% Aromatic Mixture	% Total
390	0	64.9 ± 1.2	8.3	3.0	3.8	3.5	3.7	87.4
	0.9	57.7 ± 1.9	6.9	1.9	7.2	5.3	5.8	84.7
	1.1	64.5 ± 8.6	8.5	1.9	5.2	7.7	1.4	89.3
	1.3	75.2 ± 6.7	5.4	1.1	5.1	1.5	0.4	88.6
420	0	62.3 ± 7.9	7.1	1.5	12.9	4.6	0.0	88.4
	0.9	58.9 ± 3.2	8.2	2.8	7.6	4.5	4.3	86.3
	1.1	68.5 ± 2.5	10.4	2.6	1.9	3.7	4.5	91.5
	1.3	60.6 ± 3.3	7.1	1.7	8.3	7.7	1.7	87.1

**Table 5 polymers-14-00160-t005:** Kinetic parameters used for the simulation of the pyrolysis of polystyrene.

Mechanism	Kinetic Rate Coefficient	Used Kinetic Rate Coefficients (Lmol^−1^s^−1^ or s^−1^)
Mid-chain scission	*k_b_*	1.0 × 10^−5^
End-chain scission	*k_be_*	5.1 × 10^−3^
Transfer + β-scission	*k_trb_*	1.0
Termination by combination	*k_tc_*	0–1.0
Termination by disproportionation	*k_td_*	2.0 × 10^1^
Activation of dormant species	*k_a_*	1 × 10^6^–9.0 × 10^7^ *
Deactivation of dormant species	*k_d_*	5.0 × 10^9^
Depropagation	*k_rev_*	1.1 × 10^−1^

* Value used in the simulations shown.

## Data Availability

The data presented in this study are available on request from the corresponding author.

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
