# Peer review of "Thermal Pyrolysis of Polystyrene Aided by a Nitroxide End-Functionality Improved Process and Modeling of the Full Molecular Weight Distribution"

_polymers, 2021, doi:10.3390/polym14010160_

Round 1

Reviewer 1 Report

This manuscript entitled “Thermal Pyrolysis of Polystyrene Aided by a Nitroxide End-functionality. Improved Process and Modeling of the Full Molecular Weight Distribution” by Antonio Monroy-Alonso and his coworkers focus on the process of efficient thermal pyrolysis of PS and high recovery of styrene monomer comparing with their previous works. In addition, they reported an improved model for the modeling and simulation of the process, to generate for the first time the evolution of the full molecular weight distribution without any approximation by direct integration of a large set of ODEs representing the concentrations of polymeric species during the pyrolysis process. In theory, it helps us to in-deep understand the process of efficient thermal pyrolysis of PS. In my opinion, this manuscript could be published in “Polymers” after minor revision.

  1. The part of “Polystyrene Polymerization” and “Polystyrene Pyrolysis” (including Figure 1.) could be briefly presented in main manuscript. Comparing their previous published paper (Processes, 2020, 8(4), 432.), the difference and improved operation could be emphasized in this part, the specific statement of polymerization and pyrolysis process is better to be moved into supporting information.
  2. For Figure 3. (a), b) and c)), the data of a) is repeated comparing the data of b) and c), it’s not necessary to show here. I think, a briefly statement shown in discussion part is enough.
  3. For the whole manuscript, there are still some mistakes in the format including the format of reference. Therefore, please check the whole manuscript carefully before publication.

Author Response

  1. The part of “Polystyrene Polymerization” and “Polystyrene Pyrolysis” (including Figure 1.) could be briefly presented in main manuscript. Comparing their previous published paper (Processes2020, 8(4), 432.), the difference and improved operation could be emphasized in this part, the specific statement of polymerization and pyrolysis process is better to be moved into supporting information.

Answer. We thank the Reviewer for this suggestion. We have left only brief summaries of the polymerization and pyrolysis procedures in the main manuscript and moved the detailed descriptions to the Supplementary Material as suggested by the reviewer.

As also suggested by the Reviewer and the Editor, the English has been thoroughly reviewed and polished with the help of a proficient speaker.  

2. For Figure 3. (a)b) and c)), the data of a) is repeated comparing the data of b) and c), it’s not necessary to show here. I think, a briefly statement shown in discussion part is enough.

Answer. We thank the Reviewer for this suggestion and agree with the rationale provided;  however, we prefer to keep Figure 3 (a) since it allows a direct comparison of the results at two different temperatures (390 and 420 °C). Although the data are certainly contained in Figures 3 b) and c), a direct comparison of the temperature effect is difficult if only these two figures are included. 

  1. For the whole manuscript, there are still some mistakes in the format including the format of               reference. Therefore, please check the whole manuscript carefully before publication.

Answer. Thanks to the Reviewer for the suggestion. The format of the manuscript was reviewed and corrected where needed.

Reviewer 2 Report

the paper is ready for publication in Polymers.

Author Response

We thank the reviewer for the review and final comments.

Reviewer 3 Report

In this work, the authors tackled the thermal pyrolysis process for polystyrene (PS) both experimentally and through a mathematical model. The polystyrene was synthesized via free-radical polymerization or via nitroxide mediated polymerization. Different processing conditions have been considered in terms of testing temperature, and molar ratio of nitroxide and initiator. The presence of nitroxide groups in the polymer determined a higher yield of recovered styrene and reduced overall reaction time compared to polystyrene synthesized by a free radical process. The process modelling provided the evolution of the polymer decomposition through reduction of the molecular weight as a function of time.

In my opinion, this study is well organized and fully developed, discussed in in-depth each part by highlighting the level of upgrade with respect to a previous paper by the same authors. The unique suggestion I would propose is a schematization, also simplified, of the reaction mechanisms involved in pyrolysis to have a clearer view of chemical species and bonds implicated in the thermal degradation (in the “Background: Pyrolysis Mechanism” section line 80)

Author Response

Answer. Thanks to the Reviewer for the suggestion. The suggested schematization of the reaction mechanism is already included in the Supplementary Information, but it was not cited at the point indicated by the Reviewer; therefore, we added a note at that point (line 80), so the paragraph now reads (added note in parentheses):

“…its thermal pyrolysis mechanism has not yet been completely understood but is considered to be a chain radical process that comprises the well-known free radical steps of initiation, propagation, and termination (see kinetic scheme in the Section Kinetic Mechanism in the Supplementary Material (SM))”.
